# Adiponectin/Leptin Ratio as an Index to Determine Metabolic Risk in Patients after Kidney Transplantation

**DOI:** 10.3390/medicina58111656

**Published:** 2022-11-16

**Authors:** Karol Graňák, Matej Vnučák, Monika Beliančinová, Patrícia Kleinová, Margaréta Pytliaková, Juraj Miklušica, Ivana Dedinská

**Affiliations:** 1Transplant Centre, University Hospital Martin, Kollárova 2, 036 01 Martin, Slovakia; 2Department of Internal Medicine I, Jessenius Medical Faculty, Comenius University, 036 01 Martin, Slovakia; 3Department of Gastrointestinal Internal Medicine, Jessenius Medical Faculty, Comenius University, 036 01 Martin, Slovakia; 4Department of General, Visceral and Transplant Surgery, Jessenius Medical Faculty, Comenius University, 036 01 Martin, Slovakia

**Keywords:** adiponectin/leptin ratio, kidney transplantation, posttransplant diabetes mellitus

## Abstract

*Background and Objectives:* It has been confirmed that adiponectin/leptin (A/L) ratio correlates better with cardiometabolic risk factors than hormone levels alone. The aim of our study was to determine the risk of developing post-transplant diabetes mellitus (PTDM) and other metabolic conditions depending on A/L ratio after kidney transplantation (KT). *Material and Methods*: In a prospective analysis, the studied samples were divided into three groups: control group, prediabetes and PTDM group. Pre-transplantation, at 3, 6 and 12 months after KT, we recorded basic characteristics of donor and recipient. We also monitored levels of adipocytokines and calculated A/L ratio. *Results*: During observed period, we recorded significant increase in A/L ratio in control group (*p* = 0.0013), on the contrary, a significant decrease in PTDM group (*p* = 0.0003). Using Cox regression Hazard model, we identified age at time of KT (HR 2.8226, *p* = 0.0225), triglycerides at 1 year (HR 3.5735, *p* = 0.0174) and A/L ratio < 0.5 as independent risk factors for prediabetes and PTDM 1-year post-transplant (HR 3.1724, *p* = 0.0114). *Conclusions:* This is the first study to evaluate the relationship between A/L and risk of PTDM and associated metabolic states after KT. We found out that A/L ratio <0.5 is independent risk factor for prediabetes and PTDM 1 year post-transplant.

## 1. Introduction

Post-transplant diabetes mellitus (PTDM) represents a frequent metabolic complication in kidney transplant (KT) recipients and is a serious risk factor for patient and graft survival. Even impaired glucose tolerance seems to have as significant impact on mortality after KT as PTDM itself [1,2]. According to available studies, PTDM or prediabetic condition develops in more than one third of KT recipients who have not previously suffered from diabetes [3]. In addition to glucose metabolism disorders, weight gain is almost a rule after KT, with half of the patients suffering from central obesity. Increased appetite, improved perception of tastes due to the disappearance of uremia, liberalization of dietary restrictions, as well as a sedentary lifestyle with poor overall physical condition are the factors most involved in it [4]. Central obesity is associated with hypertriglyceridemia, adipocyte-driven cytokine release, and subclinical inflammation, all of which induce insulin resistance with a high risk of PTDM development [5]. Low levels of adiponectin, which can be observed in obese patients, are closely related to insulin resistance and significantly increase the risk of developing PTDM independent of sex, age and type of immunosuppression. On the contrary, the production of leptin increases in obese patients. In our previous study, we confirmed that its increased level was significantly associated with the development of PTDM after KT [6]. Previous data also show that leptin is an independent risk factor for diseases of the cardiovascular system [7].

Despite the fact that adiponectin and leptin were independently associated with the development of metabolic syndrome (MS), diabetes mellitus (DM) type II and cardiovascular diseases, the ratio adiponectin/leptin (A/L) showed a stronger association with these pathological conditions than individual hormones [8]. A/L ratio can be considered as a marker of adipose tissue dysfunction. Due to the influence of dysfunctional adipose tissue, the amount of cardiometabolic risk markers increases, which is manifested by a decrease in the A/L ratio [9]. Its significant decrease was observed in patients with metabolic syndrome, while its decrease was correlated with an increase in the number of risk factors for MS, on the basis of which it can be considered a predictive marker of MS [9,10,11,12]. For the above reasons, the A/L ratio can represent a practical marker characterizing adipose tissue dysfunction and identify persons at increased risk of cardiometabolic diseases [13]. The results on the general population show that an A/L ratio > 1 can be considered normal, an A/L ratio of 0.5–1 indicates a moderate increase, and an A/L ratio < 0.5 a strong increase in cardiometabolic risk [14].

The aim of our study was to determine the risk of developing PTDM, prediabetic conditions and other metabolic risk factors depending on the A/L ratio in one year after KT.

## 2. Material and Methods

In our prospective study, patients actively enrolled on the waiting list for a primary KT at the Martin Transplant Center, who underwent KT during the observation period, were monitored. Patients who already had confirmed DM type I or II were not included in the follow-up. At the same time, patients with infectious complications, patients who did not undergo protocol biopsy (screening hospitalization) and those who died during the study were excluded from the study. In the third month of follow-up, screening of level of adipocytokines was performed during a short diagnostic hospitalization (graft protocol biopsy). Therefore, patients who did not undergo a protocol biopsy (poor anatomical conditions, infectious complications) were not included in the follow-up, as this would bias the results at 3 months. We divided the studied sample of patients into three groups: 1. control group, 2. patients who developed prediabetes after KT (fasting hyperglycemia, impaired glucose tolerance) and 3. patients who developed de novo PTDM after KT. In KT recipients, the initial serum level of leptin, adiponectin, interleukin 6 and 10 was measured at the time of flow cytometry crossmatch (FCXM), i.e., approximately 4 to 5 h before the procedure, and then at 3, 6 and 12 months after KT. The levels of adipocytokines and interleukins were evaluated using the ELISA method (Biomedica kits). The A/L ratio was calculated from the measured values in individual time periods. Based on the results of a previous study by Frühbeck et al. on the general population, we consider an A/L ratio above 1.0 to be normal, an A/L ratio of 0.5–1.0 indicates a medium metabolic risk, and an A/L ratio < 0.5 a high metabolic risk [14]. All participants were placed on the same immunosuppressive protocol. In the induction, antithymocyte immunoglobulin was used in a cumulative dose of 3.5 mg/kg of body weight, in the maintenance regimen tacrolimus, mycophenolic acid in a standard dosage. Regarding corticosteroids, methylprednisolone was administered at a dose of 500 mg intravenously pretransplantation and on the first day after KT, followed by a change to oral prednisone. At the time of KT, we recorded in all patients: basic characteristics of the donor (donor with extended criteria, cold ischemia time) and characteristics of the recipient (age, sex, length of dialysis treatment, underlying cause of kidney failure, delayed onset of graft function, panel of reactive antibodies, number of mismatches in class A, B, DR and DQ). In prescribed intervals after KT, we monitored risk factors for PTDM such as waist circumference, body mass index (BMI), c-peptide and immunoreactive insulin levels, lipid profile (total cholesterol, low-density—LDL and high-density—HDL cholesterol, triglycerides), vitamin D, tacrolimus level and parameters reflecting graft function such as glomerular filtration rate determined using the CKD-EPI formula (Chronic Kidney Disease—Epidemiology Collaboration Index) and quantitative proteinuria from 24 h collected urine. When diagnosing PTDM and prediabetic conditions, we used the valid criteria of the American Diabetes Association (ADA): fasting blood glucose > 126 mg/dL (7 mmol/L) in more than one case, random blood glucose > 200 mg/dL (11.1 mmol/L) with symptoms, blood glucose two hours after administration of 75 g of glucose within the oral glucose tolerance test (oGTT) > 200 mg/dL (11.1 mmol/L). The total length of follow-up was one year.

We used a certified statistical program, MedCalc version 13.1.2. (MedCalc Software VAT registration number BE 0809 344,640, Member of International Association of Statistical Computing, Ostend, Belgium). Comparisons of continuous variables between groups were carried out using parametric (*t*-test) or non-parametric (Mann–Whitney) tests; associations between categorical variables were analyzed using the χ^2^ test and Fisher’s exact test, as appropriate. Cox regression Hazard model was used for multivariate analysis for independent risk factors of PTDM in one year after KT. A *p*-value < 0.05 was considered to be statistically significant.

### Ethical Approval

All procedures involving human participants have been approved according to the ethical standards of the institutional and/or national research committee, including the 1964 Helsinki Declaration and its later amendments of comparable ethical standards.

Informed consent for included participants was checked and approved by University hospital’s and Jessenius Faculty of Medicine’s ethical committees (EK 33/2018) and all signed informed consents have been archived for at least 20 years after research completion.

The clinical and research activities being reported are consistent with the Principles of the Declaration of Istanbul as outlined in the Declaration of Istanbul on Organ Trafficking and Transplant Tourism.

## 3. Results

A total of 170 patients after primary deceased donor KT were included in the study. For known DM type I or II, a total of 28 patients were excluded; subsequently, during the follow-up, another 38 patients were excluded for other reasons (infectious complications, death, missing protocol biopsy). Thus, 104 patients were selected for prospective follow-up (Figure 1).

The level of tacrolimus was maintained in the range of 3.0 to 6.0 ng/L, and during the monitored period, we did not notice differences in its level between the studied groups. Likewise, there was no significant difference in the daily dose of prednisone.

The control group consisted of a total of 40 patients, during the monitored period, we identified a prediabetic condition in 42 patients and PTDM in 22 patients. In the individual groups, we determined the average A/L ratio at defined time intervals and compared it between them (Table 1). We found that the A/L ratio was significantly lower in the group with PTDM compared to the control group during the entire observation period, significantly lower in the group with prediabetes compared to the control group at the beginning, in the 3rd and 12th months of follow-up, and in the group with PTDM compared with prediabetes at 6 and 12 months of follow-up. Figure 2 shows the development of the A/L ratio in all three groups during the entire monitored period. During the 12 months of follow-up, the A/L ratio increased statistically significantly in the control group (*p* = 0.0013), on the other hand, it decreased significantly in the group of patients who developed PTDM (*p* = 0.0003). In the group with prediabetes, the A/L ratio also had a decreasing tendency, but this decrease did not reach statistical significance. Figure 3 illustrates the change in the distribution of individual groups according to the A/L ratio over the course of 12 months. In the control group, there was a significant decrease in patients with A/L ratio for medium risk. In the groups with prediabetes and PTDM, the changes were not statistically significant, but in both there was a decrease in the sample with a normal A/L ratio and an increase in the sample with A/L ratio for high risk.

Patients were divided then into groups according to the A/L ratio at 1 year after KT and compared them with each other in association with other recorded characteristics (Table 2). We found that patients with A/L ratio < 0.5 were in the dialysis program for a significantly longer time compared to those with A/L ratio > 1, had a higher value of BMI, waist circumference, level of insulin and triacylglycerols, worse graft function, higher prevalence of prediabetes and PTDM in 1 year after KT. When comparing with the medium risk group (A/L ratio 0.5–1), we did not notice significant differences in the incidence of prediabetes and PTDM. Compared to the normal A/L ratio, patients in this group were significantly older, had higher BMI, waist circumference and cholesterol. By comparing both risk groups, we found that patients with A/L ratio < 0.5 were significantly longer in the dialysis program, had a higher level of triacylglycerols, a lower level of HDL cholesterol, vitamin D and were younger. In the context of monitoring interleukin levels, patients at high risk (A/L ratio < 0.5) had a significantly lower level of protective IL-10 compared to the other two groups. At the same time, the group with A/L ratio > 1 showed significantly lower levels of pro-inflammatory IL-6 at 1 year after KT compared to the other two groups.

Using Cox regression Hazard model, we identified age at the time of KT ≥ 50 years (HR 2.8226, *p* = 0.0225), triglycerides level > 1.7 (HR 3.5735, *p* = 0.0174) and A/L ratio < 0.5 in 1 year after KT (HR 3.1724, *p* = 0.0114) as independent risk factors for the development of prediabetes and PTDM 1 year after KT (Table 3). At the same time, we found that pre-transplant A/L ratio was negatively correlated with BMI value (*p* = 0.0013) and A/L ratio 1 year after KT was negatively correlated with BMI (*p* = 0.0111), waist circumference (*p* = 0.0108) and triglycerides level (*p* = 0.0261) 1 year after KT. Figure 3 shows that the probability of developing PTDM and prediabetic conditions at 1 year after KT decreases significantly with increasing A/L ratio (Figure 4).

## 4. Discussion

To our knowledge, this is the first study that investigated the A/L ratio in patients after KT in the context of the risk of developing metabolic complications and cardiometabolic risk factors. In our work, we found that A/L ratio < 0.5 represents an independent risk factor for the development of PTDM and prediabetic conditions in this group of patients 1 year after KT. We identified that a higher value of the A/L ratio before transplantation, as well as 1 year after KT, statistically significantly reduced the probability of developing PTDM and prediabetes at the end of the study period. Monitoring adipose tissue hormones in association with cardiometabolic risk factors has only a short history in the transplant population, and until now these hormones have been used as separate variables. In previous studies, we confirmed that hyperleptinemia is an independent risk factor for the development of PTDM, and a low level of adiponectin was associated with insulin resistance and obesity [15].

Frühbeck et al., in a cross-sectional study in 2019 on a sample of 292 patients, evaluated the A/L ratio as a predictor of dysfunctional adipose tissue. Patients were divided into the same groups according to the A/L value (<0.5, 0.5–1, >1). In the group with high cardiometabolic risk, the authors identified significantly more patients with obesity, DM type II. and MS. At the same time, however, they state that the A/L ratio was more strongly correlated with anthropometric parameters, such as BMI, waist circumference, or body fat stores, than with parameters of metabolism and inflammation [16]. Authors Inoue et al. confirmed already in 2005 that the A/L ratio correlates with insulin resistance, even better than the Homeostatic Model Assessment (HOMA) index [17]. In our study, the A/L ratio for high risk (<0.5) was significantly associated with the incidence of prediabetes and PTDM. Since it has been shown that the probability of developing PTDM decreases significantly with an increase in the A/L ratio, its development in the post-transplantation period is therefore extremely important. We assume that over time after a successful KT, when oxidative stress, systemic inflammation subsides, immunosuppressive drug doses are reduced, and physical activity is initiated and increased, space is created for an increase in the A/L ratio. As in the above-mentioned study, in our work we also noted a significantly higher representation of obese patients in the group with A/L ratio < 0.5 compared to the group with A/L ratio > 1 expressed by BMI and waist circumference. Both of these indicators were negatively correlated with the A/L ratio, which coincides with the results in the general population [16,18].

Authors Frühbeck et al. confirmed in their study in 2017 that a low A/L ratio is an indicator of dysfunctional adipose tissue, and these patients show higher cardiometabolic risk as a result of increased systemic inflammation and oxidative stress. The authors confirmed that the levels of proinflammatory markers produced by adipose tissue, such as serum amyloid A (SAA) and c-reactive protein (CRP) strongly correlated with the A/L ratio. Increased serum concentration of these markers was associated with a lower A/L ratio. Based on these findings, they hypothesize that the A/L ratio may reflect the presence of systemic inflammation caused by adipose tissue dysfunction. It is high pro-inflammatory factors in the field of dysfunctional adipose tissue that represent mediators in the etiopathogenesis of MS [10]. For this reason, we also included IL levels in our follow-up. As one of the main pro-inflammatory cytokines, IL-6 is involved to a high degree in the activity of systemic inflammation and thus in the development of insulin resistance and DM [19]. IL-10 with its anti-inflammatory effects was identified in lower concentrations in patients with DM type II. [20]. In our studied sample, the risk group with an A/L ratio < 0.5 showed significantly lower levels of IL-10, and on the contrary, patients with a normal A/L ratio had significantly lower concentrations of IL-6 in their blood.

Both hormones, leptin and adiponectin, also contribute to the development of lipid metabolism disorders and represent a risk for fat-induced dyslipidemia [21]. We mainly identified differences in the concentrations of triacylglycerols, the level of which was significantly higher in the group with a high metabolic risk (A/L < 0.5) compared to the group with a normal A/L ratio but also with the ratio for medium risk. At the same time, patients in this group showed a low level of HDL cholesterol. This finding agrees with the conclusions of previous studies, which confirmed the negative correlation of the A/L ratio and the level of triacylglycerols [9,17]. Senkus et al. on the contrary, they detected the only significant correlation between the A/L ratio and the level of HDL cholesterol [13]. Elevated HDL cholesterol is thought to increase the secretion of adiponectin from adipose tissue, thereby promoting its anti-inflammatory and insulin-sensitizing effects [21]. In our study, the level of triacylglycerols at 1 year after KT was an independent risk factor for the development of prediabetes and PTDM. This relationship has already been confirmed earlier [22].

An interesting finding was that the monitored patients after KT with an A/L ratio for high and moderate risk of cardiometabolic complications spent a significantly longer time in the chronic hemodialysis program compared to the group with a normal A/L ratio. We consider the cause to be chronic inflammation and oxidative stress, which in long-term dialysis patients represent one of the basic aspects of their cardiovascular morbidity and mortality [23].

Limitations of our study may be the size of the investigated patient sample, the total length of follow-up and the inclusion of patients from only one Transplant center (monocentric study). Additionally, a potential limiting factor may be the lack of previous studies in this topic.

## 5. Conclusions

This is the first study to evaluate the relationship between A/L ratio and the risk of PTDM and associated metabolic states in patients after KT. In our study, we found out that A/L ratio < 0.5 is an independent risk factor for prediabetes and PTDM development 1 year after KT. The A/L ratio can be considered an indicator of adipose tissue dysfunction. The ranges of A/L ratio used by us can be a useful indicator of metabolic status and the risk of cardiometabolic complications in the post-transplantation period and could also be extrapolated to the general population. Further studies will be needed to confirm our findings.

## Figures and Tables

**Figure 1 medicina-58-01656-f001:**
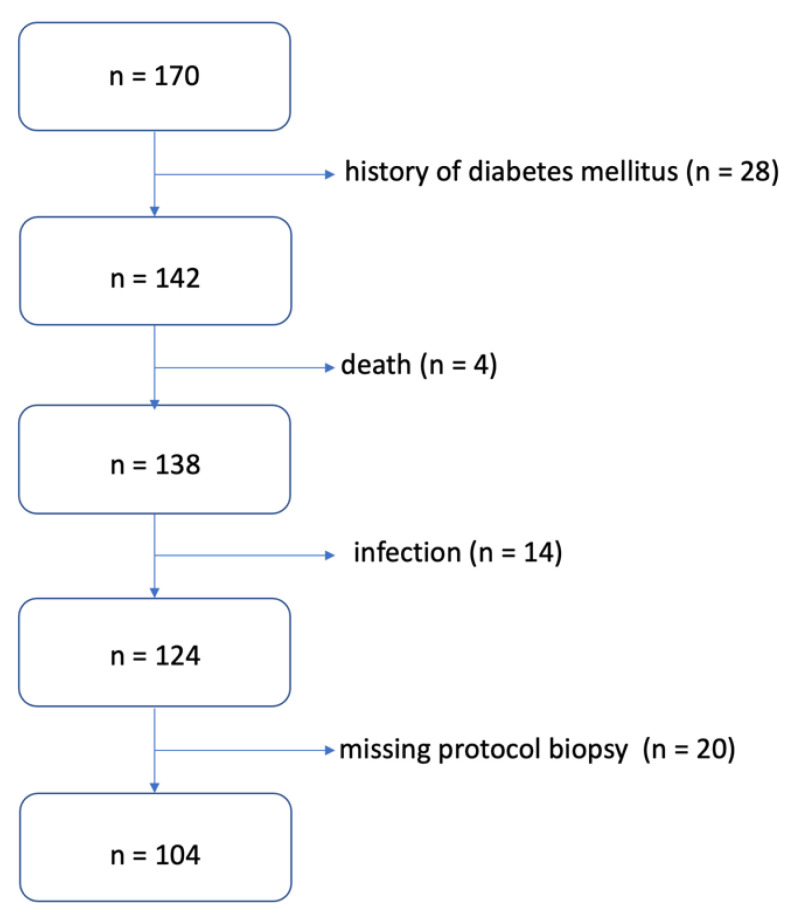
Algorithm for selecting patients for the study.

**Figure 2 medicina-58-01656-f002:**
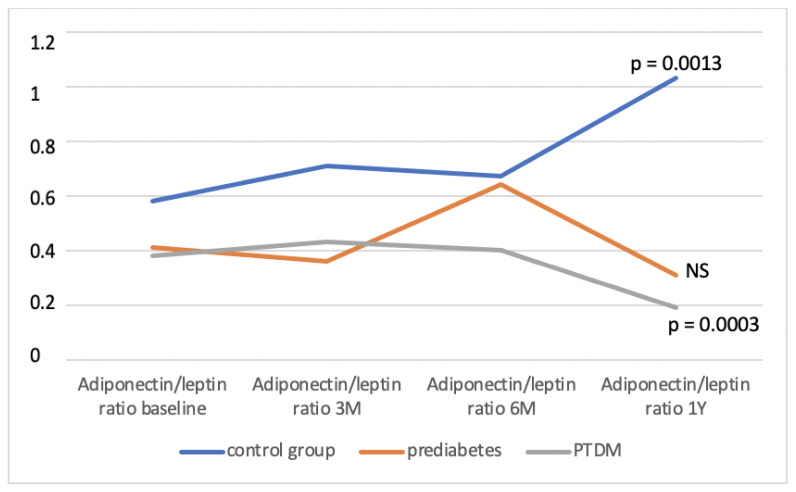
Development of the A/L ratio in observed group during the monitored period. NS—non-significant, M—month, Y—year, PTDM—post-transplant diabetes mellitus.

**Figure 3 medicina-58-01656-f003:**
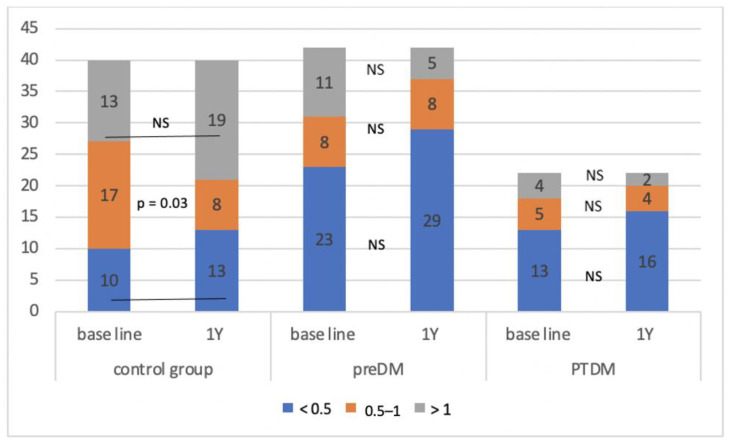
Change in the composition of the observed group according to the A/L ratio during the monitored period. NS—non-significant, Y—year, preDM—prediabetes, PTDM—post-transplant diabetes mellitus.

**Figure 4 medicina-58-01656-f004:**
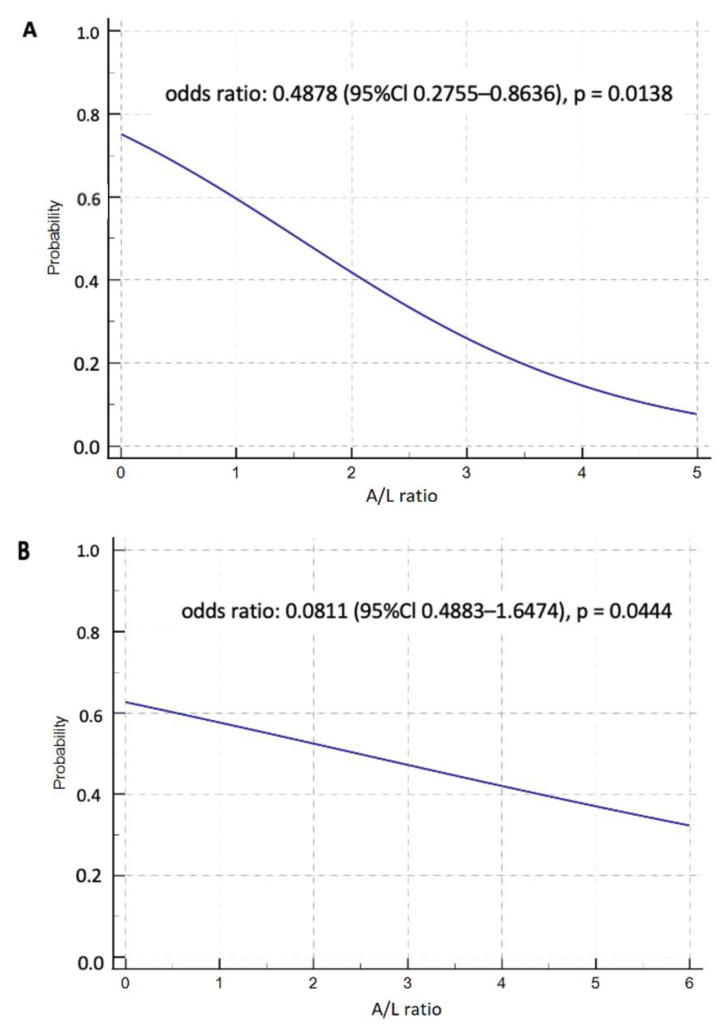
The risk of developing prediabetes and PTDM depending on the A/L ratio (**A**): baseline, (**B**): 1 year after KT. A/L—adiponectin/leptin.

**Table 1 medicina-58-01656-t001:** Comparison of observed groups based on A/L ratio.

N = 104	Control Group n = 40	Prediabetes n = 42	PTDM n = 22	*p*-Value *	*p*-Value **	*p*-Value ***
**Adiponectin/leptin ratio baseline**	0.58 ± 0.3	0.41 ± 0.4	0.38 ± 0.2	**0.0331**	**0.0069**	0.7426
**Adiponectin/leptin ratio 3M**	0.71 ± 0.6	0.36 ± 0.3	0.43 ± 0.3	**0.0012**	**0.0450**	0.3787
**Adiponectin/leptin ratio 6M**	0.67 ± 0.6	0.64 ± 0.5	0.4 ± 0.2	0.8060	**0.0455**	**0.0350**
**Adiponectin/leptin ratio 1Y**	1.03 ± 0.8	0.31 ± 0.1	0.19 ± 0.1	**<0.0001**	**<0.0001**	**<0.0001**

* control group versus prediabetes, ** control group versus PTDM, *** prediabetes versus PTDM. PTDM—post-transplant diabetes mellitus, M—month, Y—year.

**Table 2 medicina-58-01656-t002:** Comparison of the observed group according to A/L ratio in one year after kidney transplantation.

N = 104	A/L Ratio < 0.5 n = 58	A/L Ratio 0.5–1 n = 20	A/L Ratio > 1 n = 26	*p*-Value *	*p*-Value **	*p*-Value ***
**men (%)**	69 (n = 40)	60 (n = 12)	53.8 (n = 14)	0.4644	0.6755	0.1815
**Age at time of KT (Y)**	46 ± 12.7	53.9 ± 13.7	44.6 ± 15.2	**0.0213**	**0.0374**	0.6618
**Dialysis duration (M)**	62.4 ± 32.5	20 ± 16.3	18.4 ± 11	**<0.0001**	0.6932	**<0.0001**
**BMI base line (kg/m^2^)**	26.9 ± 4.4	25.1 ± 4.5	24.8 ± 4.9	0.1209	0.8322	0.0544
**BMI 1Y (kg/m^2^)**	27.3 ± 4.8	28.6 ± 3.2	23.8 ± 3.5	0.2639	**<0.0001**	**0.0013**
**Waist circumference 1Y (cm)**	103 ± 11.7	99.1 ± 14.5	86.3 ± 6.9	0.2311	**0.0003**	**<0.0001**
**C-peptid 1Y (µg/L)**	4.2 ± 5.6	3.1 ± 1.5	3.1 ± 1.9	0.3901	1.0000	0.3329
**IRI 1Y (mU/L)**	9.1 ± 3.5	8.1 ± 4.4	7.1 ± 4.2	0.3064	0.4371	**0.0256**
**Cholesterol 1Y (mmol/L)**	4.9 ± 0.8	5.1 ± 0.6	5.1 ± 1.4	0.3102	**0.0046**	0.4090
**LDL 1Y (mmol/L)**	2.9 ± 0.8	2.9 ± 0.5	3.1 ± 1.2	1.0000	0.4884	0.3700
**HDL 1Y (mmol/L)**	1.3 ± 0.4	1.7 ± 0.6	1.4 ± 0.5	**0.0012**	0.0711	0.3306
**Triglycerides 1Y (mmol/L)**	2.2 ± 1.1	1.6 ± 0.7	1.3 ± 0.5	**0.0254**	0.0969	**0.0001**
**IL-6 base line (pg/mL)**	33.2 ± 30.9	27.6 ± 20.1	21.8 ± 16	0.4523	0.2815	0.0799
**IL-6 1Y (pg/mL)**	26.2 ± 24.2	19.4 ± 9.9	12.9 ± 11.1	0.2271	**0.0415**	**0.0091**
**IL-10 base line (pg/mL)**	2.7 ± 0.9	7.7 ± 5.5	5.1 ± 4.9	**<0.0001**	0.0978	**0.0005**
**IL-10 1Y (pg/mL)**	2.8 ± 2.3	4.3 ± 1.7	7.2 ± 6.9	**0.0092**	0.0736	**<0.0001**
**Vitamin D 1Y (µg/L)**	23.4 ± 11.5	32.8 ± 17.3	30.7 ± 16.6	**0.0075**	0.6782	**0.0222**
**TAC value 1Y (ng/mL)**	6.2 ± 2.3	6.6 ± 1.9	5.8 ± 1.4	0.4867	0.1070	0.4148
**eGFR 1Y (ml/min)**	53.8 ± 22.7	54.2 ± 24.3	65 ± 16.6	0.9470	0.0803	**0.0267**
**PTDM 1Y after KT (%)**	27.6	20	7.8	0.5049	0.2293	**0.0349**
**preDM 1Y after KT (%)**	50	40	19.2	0.4429	0.1244	**0.0082**

* A/L ratio < 0.5 versus A/L ratio 0.5–1, ** A/L ratio 0.5–1 versus A/L ratio > 1, *** A/L ratio < 0.5 versus A/L ratio > 1. A/L—adiponectin/leptin, KT—kidney transplantation, BMI—body mass index, Y—year, M—month, IRI—immunoreactive insulin, LDL—low-density lipoprotein, HDL—high-density lipoprotein, IL—interleukin, TAC—tacrolimus, eGFR—estimated glomerular filtration rate, PTDM—post-transplant diabetes mellitus, preDM—prediabetes.

**Table 3 medicina-58-01656-t003:** Cox regression Hazard model.

End Point: PTDM + preDM 1Y	Hazard Ratio	95% CI	*p*-Value
**men (%)**	1.5297	0.6782–3.4502	0.3057
**Age at the time of KT ≥ 50 years**	2.8226	1.1579–6.8804	**0.0225**
**Dialysis duration ≥ 24 mesiacov**	1.4736	0.7008–3.0988	0.3066
**BMI base line ≥ 30 kg/m^2^**	1.3390	0.5315–3.3729	0.5358
**BMI 1Y ≥ 30 kg/m^2^**	0.9215	0.3288–2.5829	0.8765
**Waist circumference 1Y (men > 94 cm. women > 80 cm)**	1.2019	0.5328–2.7109	0.6577
**C-peptide 1Y > 5.19 µg/L**	1.2446	0.4905–3.1577	0.6451
**IRI 1Y > 23 mU/L**	0.8970	0.3413–2.3574	0.8255
**cholesterol 1Y > 5.17 mmol/L**	1.5615	0.6311–3.8633	0.9297
**LDL 1Y > 3.88 mmol/L**	0.8821	0.3000–2.5931	0.8196
**HDL 1Y in men < 1.03, in women < 1.29 mmol/L**	1.1066	0.4696–2.6077	0.8169
**triglycerides 1Y > 1.7 mmol/L**	3.5735	1.2504–10.2128	**0.0174**
**A/L ratio < 0.5 base line**	0.7633	0.2663–2.1880	0.6152
**A/L ratio 0.5–1 base line**	1.5126	0.5580–4.1000	0.4160
**A/L ratio > 1 base line**	0.4704	0.1776–1.2454	0.1290
**A/L ratio < 0.5 1Y**	3.1724	1.2968–7.7610	**0.0114**
**A/L ratio 0.5–1 1Y**	2.1176	0.9139–4.9071	0.0801
**A/L ratio > 1 1Y**	0.5488	0.1794–1.6781	0.2927
**vitamin D 1Y < 30 µg/L**	0.7691	0.3794–1.5591	0.4666
**TAC value 1Y > 6 ng/mL**	0.8068	0.3968–1.6404	0.5532

A/L—adiponectin/leptin, KT—kidney transplantation, BMI—body mass index, Y—year, M—month, IRI—immunoreactive insulin, LDL—low-density lipoprotein, HDL—high-density lipoprotein, TAC—tacrolimus, PTDM—post-transplant diabetes mellitus, preDM—prediabetes.

## Data Availability

The data that support the findings of this study are available from the first author upon reasonable request.

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
