# Peer review of "Adiponectin/Leptin Ratio as an Index to Determine Metabolic Risk in Patients after Kidney Transplantation"

_medicina, 2022, doi:10.3390/medicina58111656_

Round 1

Reviewer 1 Report

In this prospective study, authors attempt to determine the risk of developing PTDM, prediabetic conditions and other metabolic risk factors according to the adiponectin / leptin ratio one year after kidney transplantation. I have the following comments:

Minor comments:  

1. Introduction line 47 and line 55: re-write more clear

2. Methods line 72: patients who did not undergo protocol biopsy were excluded. Why?

3. Methods line 86: replace ..prophylactic regimen..  with…maintenance regimen..

4: Discussion line 206: replace word ..against..

5: Discussion line 240-241: re-write more clear

6: Discussion: add limitations of your study

7: Conclusion line 275-276: re-write or delete

8: Conclusion line 277: delete double dot

Author Response

Point 1: Introduction line 47 and line 55: re-write more clear

Response 1: Line 47: On the contrary, the production of leptin increases in obese patients. In our previous study, we confirmed that its increased level was significantly associated with the development of PTDM after KT. Line 55: Due to the influence of dysfunctional adipose tissue, the amount of cardiometabolic risk markers increases, which is manifested by a decrease in the A/L ratio

Point 2: Methods line 72: patients who did not undergo protocol biopsy were excluded. Why?

Response 2: In the 3. month of follow-up, screening of level of adipocytokines was performed during a short diagnostic hospitalization (graft protocol biopsy). Therefore, patients who did not undergo a protocol biopsy (poor anatomical conditions, infectious complications) were not included in the follow-up, as this would bias the results at 3 months.

Point 3, 4: Methods line 86: replace ..prophylactic regimen..  with…maintenance regimen.. Discussion line 206: replace word ..against..

Response 3, 4: Words replaced in the text.

Point 5: Discussion line 240-241: re-write more clear

Response 5: The authors confirmed that the levels of proinflammatory markers produced by adipose tissue, such as serum amyloid A (SAA) and c-reactive protein (CRP), were strongly correlated with the A/L ratio. Increased serum concentration of these markers was associated with a lower A/L ratio. Based on these findings, they hypothesize that the A/L ratio may reflect the presence of systemic inflammation caused by adipose tissue dysfunction.

Point 6: Discussion: add limitations of your study.

Response 6: Limitations of our study may be the size of the investigated patient sample, the total length of follow-up and the inclusion of patients from only one Transplant center (monocentric study). Also, a potential limiting factor may be the lack of previous studies in this topic.

Point 7: Conclusion line 275-276: re-write or delete.

Response 7: Deleted partially.

Point 8: Conclusion line 277: delete double dot

Response 8: Double dot deleted in the text.

Reviewer 2 Report

The study investigated  the correlation between adiponectin/leptin (A/L) ratio in kidney transplant patients and the risk of developing metabolic complications and cardiometabolic risk factors. The authors found that A/L ratio < 0.5 represents an independent risk factor against the development of PTDM and prediabetic conditions in this group of patients 1 year after KT. They also found that patients with a higher value of the A/L ratio before transplantation has reduced probability of developing PTDM and prediabetes at the end of the study period. Those findings make important contributions to the clinic. 

Author Response

Thank you for your review. 

Reviewer 3 Report

The study presented in the article aims to highlight the adiponectin/leptin (A/L) ratio as a predictive marker for the development of diabetes in kidney transplant patients.

The introduction highlights the mechanisms by which leptin and adiponectin are involved in the metabolic changes that precede the onset of diabetes in kidney transplant patients, as well as the values ​​at which the A/L ratio is significant for cardiometabolic risk. The patients included in the study were selected based on clearly defined criteria, the variables to be analyzed and the working hypothesis in this context were judiciously chosen and the statistical methods applied were appropriate.

The results obtained are the expression of the application of statistical tests and are illustrated in graphs and tables that clearly reflect the studied lots, the chosen variables and the statistically significant relationships between them.

The discussions are made in comparison with other studies that had the A/L ratio as a predictor of adipose tissue dysfunction.

The conclusions of the study could be more consistent, especially since the study was quite well conducted and on a sufficient number of patients so that the obtained results can be extrapolated to the general population.

Revisions of the text are still needed to eliminate possible typos. ( line 113 ethnic and probably ethical).

Author Response

Point 1: The conclusions of the study could be more consistent, especially since the study was quite well conducted and on a sufficient number of patients so that the obtained results can be extrapolated to the general population.

Response 1: This is the first study to evaluate the relationship between A/L ratio and the risk of PTDM and associated metabolic states in patients after KT. In our study, we found out that A/L ratio < 0.5 is an independent risk factor for prediabetes and PTDM development 1 year after KT. The A/L ratio can be considered as an indicator of adipose tissue dysfunction. The ranges of A/L ratio used by us can be a useful indicator of metabolic status and the risk of cardiometabolic complications in the post-transplantation period and could also be extrapolated to the general population. Further studies will be needed to confirm our findings.

Point 2: line 113 ethnic and probably ethical

Response 2: ethnical changed for ethical in the text